# Comparison between Mullite-Based and Anorthite-Based Porcelain Tiles: A Review

Kun Li [1,2,*] , Eloise de Sousa Cordeiro [3] and Agenor De Noni, Jr. [3,*]

1. School of Materials Science and Engineering, Shaanxi University of Technology, Hanzhong 723001, China
2. National and Local Joint Engineering Laboratory for Slag Comprehensive Utilization and Environmental Technology, School of Materials Science and Engineering, Shaanxi University of Technology, Hanzhong 723001, China
3. Postgraduate Program in Chemical Engineering, Federal University of Santa Catarina, Florianopolis 88040-900, SC, Brazil; eloise.ufpa@hotmail.com
* Correspondence: blueonblue@snut.edu.cn (K.L.); agenor.junior@ufsc.br (A.D.N.J.)

**Abstract:** This paper begins with an introduction to porcelain tiles. A review of the major scientific and technological features of mullite-based porcelain tiles (MPTs) and anorthite-based porcelain tiles (APTs), focusing primarily on the raw material, processing, phase evolution and mechanical behavior, is then presented. Based on the porcelain tile firing behavior and a series of physical and chemical changes that can occur, a comprehensive comparison is described. In the last part, the prospects for further developments related to MPTs and APTs are discussed.

**Keywords:** porcelain tile; mullite; anorthite; firing; mechanical behavior





## 1. Introduction

Porcelain tile is a material that offers high compact resistance and frost resistance and has good durability with low porosity [1–3]. According to ISO 10545-3 [4], its water absorption should be less than 0.5%. Porcelain tiles have been used for nearly 40 years as an excellent building material for walls, floors, pavements, and paved areas, such as squares, in towns and cities. There are four types of porcelain tile available on the market, including glazed and unglazed as well as polished and unpolished finishings. Aside from the technical performance, esthetic features are also critical for end-user decision making. The esthetic qualities of unglazed porcelain tiles are heavily influenced by the body color. The glaze layer covers the body color in the case of glazed porcelain tiles, but the body color is still important, albeit to a lesser extent than in the case of unglazed products.

Porcelain tile is a silicate ceramic that is made up of many different types of clay minerals, feldspars, and quartz [5]. It can also be classified as a triaxial ceramic due to these components [6]. With the rapid development of the porcelain industry, clay minerals, which are the most important component used in porcelain tile preparation, will soon be depleted. The major oxide components are $SiO_2$ and $Al_2O_3$, followed by $Na_2O$, $K_2O$, CaO and MgO. $Fe_2O_3$ and $TiO_2$ are common impurities, and during firing, the presence of a high content of these components will produce undesirable colors [7] in an oxidizing atmosphere. As a result, special consideration should be given to the $Fe_2O_3$ and $TiO_2$ contents. To compensate, small amounts of $ZrSiO_4$ have traditionally been used to improve the whiteness of unglazed porcelain tiles [8].

The manufacturing process used to produce porcelain tiles is well-established in the market and is similar to that of other types of ceramic tiles. Porcelain tiles differ from other classes in the following ways: raw material composition, which is more stable in terms of thermal deformations; finer particle size distribution after milling (median size 5–10 μm with 3–6% above 45 μm); higher pressing pressure (35–45 MPa); and higher firing temperature (1180–1220 °C) and time (40–60 min, cold-to-cold firing). In addition

to these distinctions, the operational window is narrow in order to maintain competitive quality, productivity, and global costs. Although the above are all conventional production parameters of porcelain tiles, the specific production parameters of porcelain tiles are not always exactly the same. To ensure that water absorption under 0.5% can be achieved, different sets of process parameters and types of unit operations can be applied.

The final microstructure of porcelain tiles is composed of a glassy matrix (55–65%) with closed pores (4–6%), dispersed crystalline particles, such as quartz, zirconite, or unmolten feldspar, and an in situ crystalized phase, such as mullite (to balance the proportions). Mullite-based porcelain tile (MPT) dominates the market and anorthite-based porcelain tile (APT) is the most promising candidate to coexist with or replace MPT. Although APT has been utilized successfully for many ceramic applications [9], including porcelain tiles [10] at a laboratory scale, it has not yet been industrial manufacturing ever since as well as MPT. As a result, a comprehensive comparison of MPT and APT is reported herein with regard to the raw materials, processing, phase evolution, and mechanical behavior based on the porcelain tile firing behavior and other process requirements.

## 2. Raw Materials

### 2.1. Mullite-Based Porcelain Tile

A typical MPT starting composition is 40–50% clay, 35–45% feldspar, and 10–15% quartz sand, as reported by Andreola and co-workers [11]. In particular, the formula containing 50% kaolinitic clay, 40% feldspar, and 10% quartz sand has been studied in-depth by Romero and co-workers, with a focus on, but not limited to, the kinetic, microstructural, and mechanical behavior of porcelain tiles [12–18]. In this formula, there are three main oxides: $K_2O$, $Al_2O_3$, and $SiO_2$, mainly introduced by feldspar, clay, and quartz, respectively, while other oxides are added as low-content impurities of these three raw minerals. Due to the three main components, compositions are mostly members of the $K_2O$-$Al_2O_3$-$SiO_2$ system.

In general, kaolin, china clay [19], illite clay [20], pyrophyllite [21], and flint clay are used as clay minerals to provide plasticity and dry mechanical strength to the green body. Clay minerals, including calcined kaolin (containing $Al_2O_3$), and flux components, are used to develop the mullite and glassy phase. Microcline [22], albite [23], pegmatite [24], perlite, nepheline syenite [25], spodumene [26], diopside [27], and tremolite are used as feldspar minerals. Feldspar minerals produce a liquid phase at low temperatures (the theoretical melting points for microcline and albite are around 1250 °C and 1100 °C, respectively) to promote densification of the porcelain tile. The use of these raw materials leads to a major liquid phase belonging to the $Na_2O$-$K_2O$-$Al_2O_3$-$SiO_2$ system. Talc and calcite can be used as minor fractions, up to 5%, to form additional eutectic phase with feldspar and reducing the firing temperature [28]. Sands are the most commonly used quartz minerals, but quartz is also present in industrial raw materials, such as clays and feldspars. These improve the thermal and dimensional stability due to a high melting point, which partially offsets firing shrinkage. High representative oxide content, low impurity oxides, and low loss on ignition (LOI) are the main criteria used to evaluate whether a raw mineral material is of high quality. Clay minerals with high $Al_2O_3$ content, feldspar minerals with high $K_2O$ and $Na_2O$ content and near-zero LOI, and quartz minerals with almost 100% $SiO_2$ are all ideal minerals for MPT preparation. High-grade minerals will provide MPT materials with low water absorption, high density, lower firing shrinkage, and enhanced mechanical properties.

With the rapid depletion of natural mineral resources, alternative compositions for porcelain tile preparation urgently need to be developed. In this regard, two new ways to prepare alternative porcelain tiles have been developed. One is to add a green body enhancer [29,30] to produce porcelain tiles with less or no plastic minerals, and the other is to incorporate bulk urban/industrial waste [5,6,11,31–37] in the porcelain tile manufacturing process. It is worth noting that for porcelain tiles prepared with less or no plastic minerals (such as clay mineral) the phase composition/ratio will differ significantly from that of traditional MPTs. Producing porcelain tiles with bulk urban/industrial waste is a promising approach due to the bulk stock and considerable proportion of useful ingredients.

In addition, reusing waste has beneficial effects on both the environment and the economy. Future approaches to porcelain tile production with less/no clay minerals should also aim to provide a decorative appearance, large size, better mechanical properties, lower firing temperature, and shorter firing time in order to reduce energy consumption and carbon dioxide emissions.

### 2.2. Anorthite-Based Porcelain Tile

The composition of an APT body can comprise around 20% clay mineral, 25% wollastonite, 30% alumina, 20% quartz, and 5% basic magnesium carbonate [38]. Anorthite-based porcelain tile and hard porcelain have been studied by Capoglu and coworkers [38–41]. Their compositions are made up of three main oxides, $SiO_2$, $Al_2O_3$, and CaO, and thus, they belong to the $CaO-Al_2O_3-SiO_2$ system. These oxides are supplied by wollastonite, clay mineral, alumina, and quartz. MgO has been used a sintering aid [42] to produce fired materials and enable their easy vitrification and densification. Furthermore, investigations have been carried out suggesting that wollastonite is the best CaO source considering $Ca(OH)_2$, $CaCO_3$, marble powder, gypsum mold waste, dolomite, wollastonite, and calcite [43,44]. However, Ibañez and Sandoval reported the relatively low availability of natural wollastonite deposits [45]. The same authors cited other papers that have addressed applications of wollastonite-clay-feldspar mixtures for high- and low-porosity ceramic products but not porcelain tiles. Wollastonite can reduce the drying and firing shrinkage and improve the dried and fired mechanical strength, but it narrows the sintering interval [45,46]. In recent years, new wollastonite deposits have been explored to supply a superior low-temperature fast-firing mineral, attracting considerable attention not only from academic researchers but also the ceramics industry.

The clay minerals and quartz minerals used in APT preparation are the same as those used for MPT. Wollastonite and calcite have been used as calcium-containing minerals. It can be inferred from the description above that clay minerals with high $Al_2O_3$ content, minerals with high CaO content and near-zero LOI, and quartz minerals with almost 100% $SiO_2$ all are ideal minerals for APT preparation. High-grade minerals will provide APT with low water absorption, high density, lower firing shrinkage, and enhanced mechanical properties, as described in the section on MPT. APT was successfully developed by Tai et al. [47] using aluminous cement as a binder instead of traditional plastic raw materials. Wu et al. [48] used diopside and magnesia clay as a source of MgO and CaO. Tarhan [49] added 12% of a diopside-based frit to a porcelain tile composition. In both cases, anorthite and diopside were crystallized in the final body. Selli [8] added 1.0–1.5% of CaO and 4–6% of $Al_2O_3$ to a regular porcelain tile composition and obtained anorthite and mullite as crystallized phases. In these cases, the reference system was $CaO-MgO-Al_2O_3-SiO_2$.

Another way to describe or analyze a porcelain tile composition is with regard to the oxide components. It has been noted that the content of $SiO_2$ should be 60–70%. When the $SiO_2$ content is in excess of 75% the porcelain tile can crack due to the transformation of β-$SiO_2$ to α-$SiO_2$ in the quartz [50] during cooling. The appropriate $Al_2O_3$ content is above 20%. However, production experiences have shown that increasing the $Al_2O_3$ content in the formula can increase the firing temperature and expand the sintering temperature range. With an $Al_2O_3$ content of less than 18%, although the porcelain can be fired, the sintering temperature range is narrow, and product deformation can easily occur. In the MPT formula, mullite is derived from the clay/feldspar components [51]. If the $K_2O$ ($Na_2O$) content in the formula is too low (less than 5%), the product will be difficult to sinter, while a content that is too high (more than 10%) will significantly reduce the firing temperature, and product deformation can easily occur. It has been verified that too much CaO (exceeding 15%) will significantly narrow the firing temperature range, resulting in difficulties during industrial production.

Based on the scientific research, from the economical perspective, the lower the variety of raw minerals used in the composition of porcelain tiles the better. In addition, a large number of components makes analysis difficult to carry out. From the perspective of engineering

applications, multi-variety of raw minerals used in the formulation of porcelain tiles is preferable. Also, when more types are used, the content of each is smaller. The raw minerals can be replaced with similar minerals, and the properties of the product will remain almost unchanged, thereby significantly reducing the limitation of mineral resource types.

## 3. Processing

Porcelain tile production lies within the category of powder technology processes. The milling process can be dry or wet and extrusion or pressing can be used as a forming stage. Tunnel or roller kilns can be used for firing. Despite other pathways being available, wet milling, pressing and roller kiln are the most common processes and they are widely used worldwide. Wet milling, as well as powder pressing, can be used in batch or continuous processes. Batch firing is mainly limited to laboratory size, whereas continuous and fast firing are more widely used due to productivity and cost considerations. The type of product (glazed or unglazed, small or large, thin or thick) and the target market (high or low added value; residential, commercial or industrial applications) determine the section of the process options. Also, the availability of raw materials can sometimes be a deciding factor. Unglazed porcelain tiles, for example, cannot be made with low-whiteness raw materials. In all circumstances, firing is the main step.

The choice of dry versus wet milling is one of the most crucial aspects of establishing the flow chart for a new plant. Both processes can create glazed porcelain tiles that are nearly identical [52]. When taking the dry route, the main objective is usually to obtain a low-cost product. However, the milling technology is only one factor to consider and other factors of the product design, such as the quality of the raw materials and the glazed surface, also need to be taken into account.

Dry-processing is the simplest method available to produce porcelain tiles. Some authors show that this approach is less energy-intensive than the wet route [53–55]. Ball or pendulum mills are used to blend and mill the dried raw materials. Following milling, the mixes are subjected to a particle size-enlargement process, for instance, using as a muller-mixer, to granulate the powder by adding 9% to 11% moisture [52].

The wet-processing route of obtaining the granulated powder composition is more expensive. Ball milling is used to reduce the size of the particles in the water suspension in this case. A typical solid load in the slurry is 60% to 65% by weight. Inorganic or organic ionic dispersants are commonly employed to produce the highest possible solids loading in order to save energy in the spray-drying process. Spray-dried powder is used to dry and granulate the particles at the same time. When compared to the dry route, the wet route produces more homogenous mixing between particles. Spray-dried powder is well-rounded, with a moisture content between 5.5% and 7.5%.

The granulates are pressed onto a green porcelain tile body and then fired. The porcelain tile can then be cut, edged, polished and lastly packaged as a saleable product. In terms of processing route, there are no substantial distinctions between MPT and APT.

The sintering of porcelain tiles is mainly attributed to the diffusion behavior driven by thermodynamics. During sintering, both solid-phase mass transfer and the formation of a high-temperature liquid phase are conducive to particle rearrangement, filling open pores and completing the densification of sintering. Seen from its appearance changes, the porcelain body exhibits volume shrinkage, a decrease in open porosity and an increase in both strength and bulk density during sintering.

## 4. Phase Evolution

To better understand the differences between MPT and APT, a comparison of chemical compositions of green bodies was given in Table 1. APT green body has higher $Al_2O_3$, extremely higher CaO, MgO and lower $SiO_2$ than MPT's; both green bodies contain kaolin and quartz phases, besides MPT green body comprises feldspar while APT green body comprise wollastonite and corundum phases. Variances in the source components and their contents of MPT and APT greatly effects their firing behavior, phase evolution and

eventually determines the final microstructure and mechanical properties. The final microstructure and properties of ceramic materials are established during firing, which is the most important step in their processing. The main reaction is the densification through-out sintering. Many other reactions occur in addition to sintering, including clay mineral and other hydroxide dehydroxylation, organic matter volatilization and oxidation, phase transitions, thermal expansion, melting and crystallization [56]. In the case of porcelain tile manufacturing, liquid phase sintering is usually the dominant mechanism. Distinct final phase kind, content and distribution, which is determined directly by the raw materials kind, batch formula and processing procedure feature, will significantly affect the mechanical properties of final product. Hereby, we investigated the final phase composition of two kinds of porcelain tile, to understand and interpret the mechanical property differences (namely bending strength) between them.

**Table 1.** Typical chemical compositions of MPT and APT green bodies (From the authors).

| Name | SiO$_2$ | Al$_2$O$_3$ | Fe$_2$O$_3$ | TiO$_2$ | CaO | MgO | K$_2$O | Na$_2$O | I.L | Major Phase |
|------|---------|-------------|-------------|---------|-----|-----|--------|---------|-----|-------------|
| MPT | 65.48 | 22.06 | 0.50 | 0.28 | 0.31 | 0.23 | 2.6 | 3.37 | 5.17 | Kaolin, feldspar, Quartz |
| APT | 42.59 | 37.77 | 0.24 | 0.21 | 11.27 | 2.49 | 0.41 | 0.40 | 4.62 | Wollastonite, Kaolin, Corundum, Quartz |

### 4.1. Mullite-Based Porcelain Tile

The sintering temperature of MPT is around 1200 °C. With the elevation of sintering temperature, a glassy matrix is gradually formed by the melting of potassium-rich clays and alkaline-rich rocks such as feldspars at around 1100 °C. Residual feldspar particles can be present in the final microstructure depending on the process time and temperature. Quartz is present in the amounts of 20–25% as α-quartz and the particles remain unreacted, with the exception of those below 6 μm, which can be partially or completely dissolved by the liquid phase. The main reaction involving quartz is its allotropic transition β→α at 573 °C during the cooling step. This transition generates thermal stress and can collapse the body if the cooling rate is too fast (cooling stage, around 573 °C, less than 30 °C/min). Mullite, after quartz, is the second richest crystalline phase in the porcelain tile microstructure (7–16%). In some cases, small amounts of corundum or zirconite are also found [57]. Mullite phase can be presented in two stoichiometric forms [58], 3Al$_2$O$_3$·2SiO$_2$ or 2Al$_2$O$_3$·SiO$_2$, and in MPT, it usually stands for the former. In general, mullite is developed in a routine MPT sintering process with the following steps [59]: (a) dehydroxylation of hydroxyl groups contained in kaolinite (Al$_2$O$_3$·2SiO$_2$·2H$_2$O), introduced by clay or kaolin, to form metakaolin (Al$_2$O$_3$·2SiO$_2$) at around 550 °C; (b) metakaolin decomposition to form a spinel-type structure (Al$_8$(Al$_{13.33}$⊕$_{2.66}$)O$_{32}$ or Si$_8$(Al$_{10.67}$⊕$_{5.33}$)O$_{32}$), while amorphous free silica is released at around 950–1000 °C; and (c) the spinel phase transforms to mullite phase at above 975 °C. Notably, a MPT body comprises two kinds of mullite: primary mullite and secondary mullite. Primary mullite is comprised of very small crystals (with low aspect ratio, approximately 1–3:1) derived from clay at a lower temperature [60]; secondary mullite consists of needle−like crystals (with high aspect ratio, from 5:1 to 20:1) and grows through the liquid phase while primary mullite serves as a seed for the secondary mullite crystallization [16,61]. In the K$_2$O-Al$_2$O$_3$-SiO$_2$ system, primary mullite forms at around 985 °C [62] under a fast rection. Secondary mullite formation is much slower in comparison with primary mullite and its completion is dependent on the process time and temperature. In a typical porcelain tile microstructure, the primary mullite accounts for nearly all of the mullite [16,28].

### 4.2. Anorthite-Based Porcelain Tile

In the case of APT, anorthite is the main phase with a representative composition of around 52% anorthite, 12% corundum, 8% cristobalite, and 28% glassy phases. Anorthite is formed from the CaO introduced by wollastonite, Al$_2$O$_3$ introduced by clay minerals and/or alumina, and SiO$_2$ introduced by quartz [38]. The formation of anorthite begins



with the dehydroxylation of kaolinite ($Al_2O_3 \cdot 2SiO_2 \cdot 2H_2O$, introduced by clay minerals) to form metakaolin ($Al_2O_3 \cdot 2SiO_2$) and ends with the reaction of metakaolin ($Al_2O_3 \cdot 2SiO_2$) and wollastonite ($CaO \cdot SiO_2$). The formation temperature of anorthite is around 1000 °C [63] in the $CaO$-$Al_2O_3$-$SiO_2$ system. In addition, anorthite can also be formed from metakaolin and CaO [64], or mullite, $SiO_2$, and CaO [65].

The firing time appears to affect the form of the anorthite phase with a rounded shape, or low aspect ratio, for a short time [49] and needle-like crystals for a longer time, showing similar firing behavior as the mullite phase. The anorthite phase is a congruent melting compound in the $CaO$-$Al_2O_3$-$SiO_2$ system, just the same as the mullite phase in the $K_2O$-$Al_2O_3$-$SiO_2$ system. More specifically, they show a preferred orientation of c-axis growth under constant heating for a relatively long time. It is worth noting that the formation velocity of the anorthite phase in a typical APT body is between those of the primary and secondary mullite phases in a typical MPT body under the same processing condition. The anorthite phase formation velocity is competitive with both types of mullite, especially secondary mullite. This point is of interest since the $K_2O$-$Al_2O_3$-$SiO_2$ and $CaO$-$Al_2O_3$-$SiO_2$ systems could be joined together with the final phase of primary mullite, the glassy phase, and the anorthite phase (to replace secondary mullite). This new combination could probably lead to a new mullite−anorthite porcelain with distinct performance compared to MPT or APT, and it could achieve some unexpected properties. Attempts to determine the raw composition should be made since the processes of anorthite and mullite formation are competitive reactions. Brasileiro et al. [66] reported a reduction in mullite content from 7.7% to around 3.8% when wollastonite or diopside were added to an industrial MPT composition.

For these two types of porcelain tile, the phase compositions were determined by the types of raw material and their contents, indicating different ternary systems ($K_2O$-$Al_2O_3$-$SiO_2$ or $CaO$-$Al_2O_3$-$SiO_2$). Indeed, the main components (e.g., oxide) in a formula determine the peak firing temperature, firing temperature range, and firing behavior. The firing temperature was mainly defined by the content of $Al_2O_3$, and its particle diameter was affected by the proportion of oxide flux. Although $Fe_2O_3$ and $TiO_2$ are not expected to be present in a green porcelain body, they will promote sintering and densification to some extent.

A comparison of the mullite development in MPT and anorthite development in APT is given in Figure 1. For mullite formation, kaolin minerals are the most critical, and for the formation of anorthite, kaolin and wollastonite minerals are both indispensable.

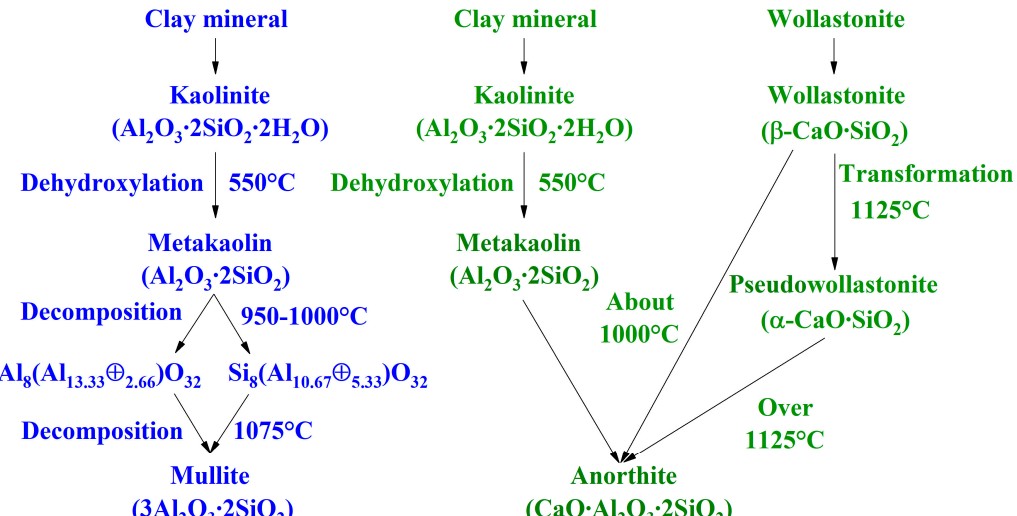

**Figure 1.** Comparison of mullite (blue) development in MPT with anorthite (green) development in APT.

Table 2 shows a comparison of the fundamental physical properties of typical crystal phases that could form in a porcelain material [67–79]. It can be seen that the corundum phase has much higher thermal conductivity (nearly 6 times that of mullite). This indicates

that if a material with high thermal conductivity is prepared, corundum phase is far more suitable than mullite phase as a target phase.

**Table 2.** Comparison of fundamental physical properties of typical crystal phases formed in porcelain materials.

| Phase | Chemical Composition | Density (g/cm$^3$) | Refractive Index | Thermal Conductivity ([W/(m·K)]/K) | Volume Thermal Expansion Coefficient ($10^{-6}$/°C)/(°C) | Melting Point (°C) |
|---|---|---|---|---|---|---|
| $\alpha$-Quartz | $SiO_2$ | 2.53 | 1.54 | 3.7–14.0/500–800 [67] | 23.8–86.0/298–773 [68] | 1710 |
| $\beta$-Quartz | $SiO_2$ | 2.65 | 1.54 | 4.1–4.8/900–1100 [67] | nearly 0/575–1100 [68] | 1710 |
| Mullite | $3Al_2O_3 \cdot 2SiO_2$ | 3.16 | 1.64 [49] | 6.0 | ~16.7/300–900 [69] | 1850 |
| Corundum | $Al_2O_3$ | 3.95 | 1.76 | 35 | 22.9–32.4/20–2025 [70] | 2050 |
| Albite | $Na_2O \cdot Al_2O_3 \cdot 6SiO_2$ | 2.61 | 1.53 | 2.3 [71] | 21.6–37.3/25–1200 [72] | 1118 |
| Microcline | $K_2O \cdot Al_2O_3 \cdot 6SiO_2$ | 2.54 | 1.52 | 2.4 [71] | ~13/100–1000 [73] | 1290 |
| Leucite | $K_2O \cdot Al_2O_3 \cdot 4SiO_2$ | 2.45 | 1.51 | 1.1 [71] | 22–30/25–1000 [74] | 1120 |
| Anorthite | $CaO \cdot Al_2O_3 \cdot 2SiO_2$ | 2.75 [75] | 1.58 [75] | 1.7 [71] | 14–60/200–900 [76] | 1550 [77] |
| Cordierite | $2MgO \cdot 2Al_2O_3 \cdot 5SiO_2$ | 2.61 | 1.54 | 2.7 [71] | 2.6/25–600 [78] | 1460 |
| Diopside | $CaO \cdot MgO \cdot 2SiO_2$ | 3.27 | 1.68 | 5.6 [71] | 33.3/24–1000 [79] | 1330 [77] |

## 5. Mechanical Behavior

Vickers microhardness (HV), bending strength (σf), fracture toughness (KIC) and Young's modulus (E) are all mechanical properties of porcelain tiles that are closely related to the phase composition and microstructure features [80]. The bending strength, along with the water absorption (Wa), are key aspects for determining the quality of porcelain tiles.

The mechanical properties of all varieties of porcelain tile are influenced by the porosity (P), which is one of the most important microstructural parameters [81]. Low porosity increases the E value and decreases the flaw size, which increases the KIC and σf values. The mechanical characteristics improved with interactions between the processing parameters, which led to reduced porosity [57]. The interactions among the particle size distribution, moisture, and forming pressure promote sintering due to the improved green density. Finer particles increase the sintering rate, especially if the particle packaging [82] is not reduced significantly, reducing the final porosity, pore size, and flaw size. The most common pores in porcelain tiles can be divided into two types: connected open pores with irregular shapes and closed pores with regular shapes (usually round and oval).

The chemical and mineralogical composition of porcelain tiles have a strong effect on the mechanical properties. A component that increases the sintering rate and thus lowers the porosity and pore size distribution will also improve the mechanical properties. Other effects are dependent on the dispersed crystalline particles and clusters that remain or form during the sintering. The role of such components are the key aspects that differentiate mullite from anorthite porcelain tiles. In general, high-strength porcelain tiles have the following characteristics: few or no open/connected pores, a small number of regular-shaped closed pores, dense cross-sections without obvious cracks or defects, and a crystal phase that is tightly surrounded by the glass phase or matrix. The frequency histogram in Figure 2 shows bending strength data taken from the literature on MPT and APT [10,14,22,28,38,40,44,47,75,78,80,81,83–88]. In many cases the values are similar, but APT appears to have higher bending strength values than MPT.

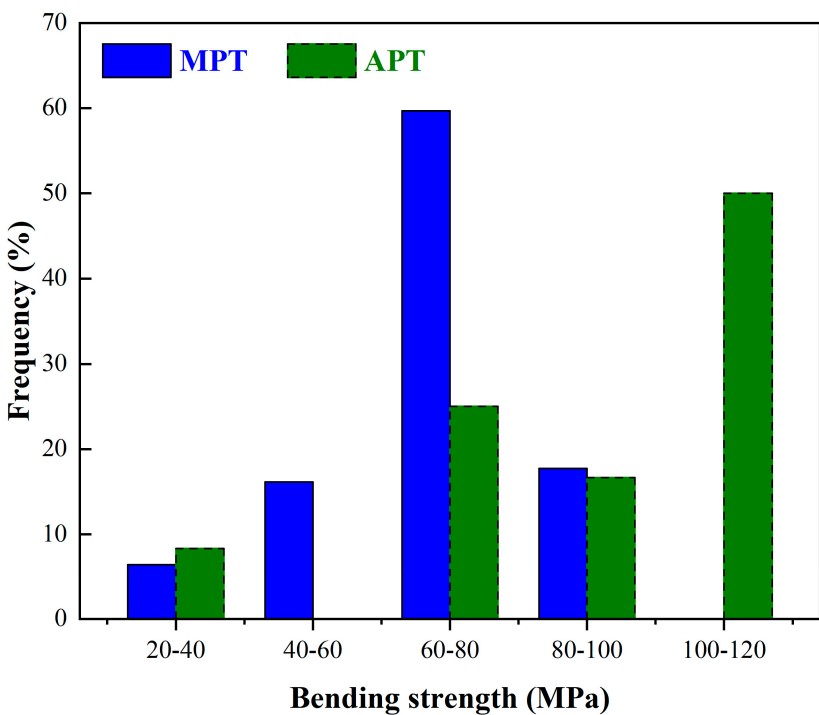

**Figure 2.** Frequency histogram for bending strength of porcelain tile samples, data from the literatures [10,14,22,28,38,40,44,47,75,78,80,81,83–88]. Sample number for MPT = 62, APT = 12.

*5.1. Mullite-Based Porcelain Tile*

The crack deflection and glass matrix stress effect are the major reinforcement mechanisms in porcelain tiles. Every dispersed crystalline phase or particle can cause crack deflection, enhancing the fracture toughness. Finer particles, when compared to coarser particles, have a more beneficial effect because they result in a shorter mean free path between particles [89]. Finer particles can help reduce the flaw size, improving the bending strength. This is one of the beneficial effects of secondary and primary mullite clusters. The holding time is usually inadequate (usually 10–15 min) to form a significant amount of interconnected secondary mullite. The same beneficial effects can be achieved with finer quartz particles [90]. Coarse quartz particles can form clusters and increase the flaw size while reducing the bending strength [91].

According to Selsing's model [92], particles with thermal expansion mismatch generate microscopic residual stress due to glass matrix stress effects. Quartz particles have a higher thermal expansion coefficient (TEC) than the glass matrix, causing tangential compressive stress across the matrix. As a result, the fracture energy and the bending strength increase [93]. The size of the quartz particles affects (positively or negatively) the mechanical properties. Particles larger than ~45 μm are surrounded by microcracks that have detached from the matrix and are unable to generate stress in the matrix. The Young's modulus [94] and bending strength can both be reduced by the cracks. Microcracks partially surround particles above ~6 μm and below ~45 μm, causing compressive stress in the glass matrix. Particles below ~6 μm are partially dissolved and generate tensile stress, which lowers the fracture energy by forming a glass silica-rich interface with a lower TEC than the glass matrix [93]. Mullite also has a lower TEC than the glass matrix, which reduces the fracture energy.

The dispersed crystalline phase and particles in mullite porcelain tiles have a notable antagonistic effect on the mechanical properties. In addition to this complex relationship, the product is subjected to a fast cooling rate, which results in macroscopical residual stress, increasing the bending strength [95]. A fast cooling rate can also have a negative impact on the mechanical characteristics, increasing the flaw size [95].

Finding a balance between the positive and negative effects of each reinforcing mechanism is a key factor in relation to improving the mechanical properties. The data indicate that the bending strength of mullite porcelain tiles could be increased from ~60 MPa to ~90 MPa using this approach [28,96].

### 5.2. Anorthite-Based Porcelain Tile

The crystalline phase type, the content of each phase, and the microstructure of the anorthite-based porcelain tile show the same behavior when compared to the mullite-based porcelain tile. However, in contrast to MPT, the bending strength of APT is governed by the anorthite content, glassy phase, reinforcing phase (such as corundum and zirconite) and open porosity and is affected by the prestress obtained and blisters.

The most striking difference between MPT and APT is the representative phase formed as well as the way it is distributed. In the literature, APT reportedly has a higher strength than MPT, and this difference probably originates from the ratio of crystalline to amorphous phase, which, for MPT is 2:4 (~55 MPa), and for APT, it is close to 3:1 (~110 MPa). A high content of CaO will greatly narrow the firing temperature range, and overfired deformation will be observed. The MPT body contains $K_2O$ and $Na_2O$, while the APT body contains CaO, which will form a soluble calcium glass phase at high temperature, significantly decreasing the viscosity of the melt and leading to a much narrower firing temperature range. Thus, the CaO content should be strictly controlled. During sintering, the glassy phase formed with CaO has lower viscosity than that of $K_2O$ or $Na_2O$, leading to easier particle rearrangement and the filling in of pores, thus accelerating the densification process. Relatively low viscosity means less open porosity and less glassy phase, leading to high mechanical strength. On the other hand, an accelerated densification process shortens the period from melting to completion and narrows the acceptable temperature range.

Figure 3 shows the typical microstructures of MPT and APT. In the case of MPT, the micrograph is of an industrial sample. The cross-section was polished with 2 μm alumina paste. No sputtering of a conducting element was applied. The SEM (Philips XL30 CP, Amsterdam, The Netherlands) setup was: 20 kV, spot 5.8, BSE detector, 0.2 mBar, and 800× magnification. The original photo was adjusted by balancing the brightness, contrast, and sharpness using image editing software.

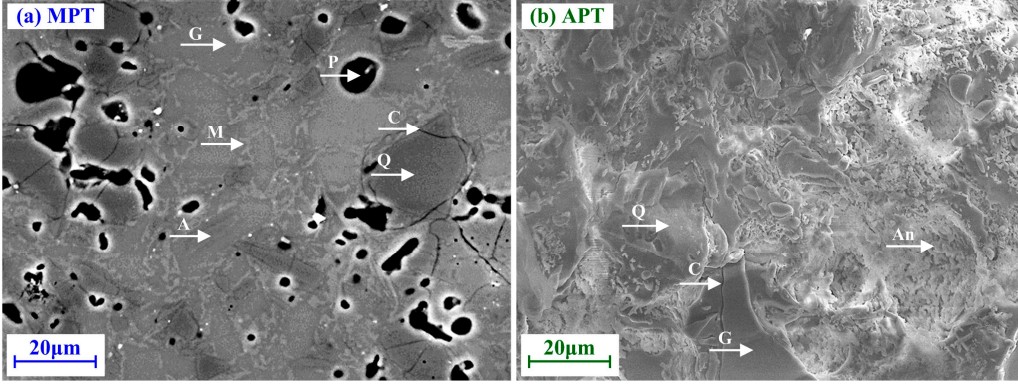

**Figure 3.** SEM of polished cross-section of MPT and fractured surface of APT. G, glass matrix; P, closed pore; Q, quartz particle; C, cracks; A, unmelted albite; M, primary mullite cluster; An, anorthite cluster.

In the case of APT, the micrograph is of a laboratory sample. The fractured surface was subjected to Au sputtering, without polishing or etching.

The MPT microstructure was formed by a glass matrix (G), closed pores (P), and quartz particles (Q) and surrounded by cracks (C), unmelted albite (A), and closed pores (P). Primary mullite clusters (M) can be observed surrounding the quartz particles and the glass matrix originated from albite particle melting. In this case, these cluster do not appear to be interconnected because of the low amount of kaolinite in the starting composition [82]. The APT microstructure was formed by interconnected anorthite clusters embedded in a

glass matrix (G), surrounded by quartz particles (Q) and cracks (C). The absence of irregular pores and closed pores was probably due to a relatively long firing cycle, which led to the redistribution of particles and the elimination of pores. The criss-crossing anorthite and surrounding quartz phases probably contribute to the strength of the fired body.

A comprehensive comparison of MPT and APT is given in Table 3, based on the above discussions.

**Table 3.** Comprehensive comparison of MPT and APT.

| | Items | MPT | APT |
|---|---|---|---|
| **Composition** | Main mineralogical components | Clay, feldspar, quartz | Wollastonite, clay, alumina, quartz |
| | Main oxides components | $SiO_2$, $Al_2O_3$, $K_2O$, $Na_2O$ | $SiO_2$, $Al_2O_3$, CaO, MgO |
| | Phase diagram attribution | $SiO_2$-$Al_2O_3$-$K_2O$ ternary system | $SiO_2$-$Al_2O_3$-CaO ternary system |
| **Processing** | Particle size | d50 5–6 μm, Fine/Coarse is 1:10 | |
| | Powder moisture | 5–7% | |
| | Forming pressure | 35–45 MPa | |
| | Sintering temperature | 1180–1220 °C | 1120–1230 °C |
| | Sintering temperature range | Wide (40 °C or wider) | Narrow (30 °C or narrower) |
| | Holding time | 4–6 min | >15 min |
| | Cold-to-cold time | 35–60 min | |
| **Final composition and properties** | Phase composition | 20–25% α-quartz, 12–16% mullite, balanced by amorphous phase | 52% anorthite, 12% corundum, 8% cristobalite, and 28% glassy phases |
| | Microstructure | Glassy matrix embedded with quartz, mullite phase; secondary mullite acts as reinforcing phase | Glassy matrix embedded with anorthite phase; corundum phase acts as reinforcing phase |
| | Ratio crystalline: amorphous phases | ~1:2 | ~3:1 |
| | Mechanical strength | ~55 MPa | ~110 MPa |
| | Whiteness | ~80 | ~92 |
| | Total evaluation | Wider sintering temperature range, medium strength, medium comprehensive performances, small- to large-scale production | Narrower sintering temperature range, good strength: good comprehensive performances, still restricted in large-scale production |

## 6. Prospects and Outlook

The raw material, processing, phase evolution, and mechanical behavior of MPT and APT were discussed. The conclusions could be draw as follows:

1.  Typically, APT is prepared using 50% clay, 40% feldspar, and 10% quartz, and it can be attributed to the $SiO_2$-$Al_2O_3$-$K_2O$ ternary system; an MPT can be prepared using 20% clay mineral, 25% wollastonite, 30% alumina, 20% quartz, and 5% basic magnesium carbonate, and it can be attributed to the $SiO_2$-$Al_2O_3$-CaO ternary system. Variances in the source components and their contents of MPT and APT greatly effects their firing behavior and phase evolution and eventually determines the final microstructure and mechanical properties. The insufficient reserves of wollastonite in major porcelain tile manufacturing countries affects the industrial application of APT.

2.  MPT and APT have no substantial distinctions in their processing routes except the sintering temperature, sintering temperature range, and holding time. The mature similar parameters are a mean powder particle size of 5–6 μm, 10% fine particles and 90%, coarse particles, 5–7% granulating powder moisture, forming pressure of 35–45 MPa, and cold-to-cold time of 35–60 min. The average sintering temperature of APT is 40 °C lower than that of MPT, whereas its sintering temperature range is more than 40 °C narrower than that of MPT. A much narrow sintering temperature range is the main obstacle the industrial application of APT. A combined system of the $SiO_2$-$Al_2O_3$-$K_2O$ and $SiO_2$-$Al_2O_3$-CaO ternary systems, as well as flux consisting of

both feldspar and a magnesia-containing component, may improve the firing behavior and further promote the industrial application of APT.

3.   Mullite is the feature phase in MPT, and anorthite is the feature phase in APT. Due to a larger ratio of crystalline to amorphous phase, the crystalline phase type, and a higher CaO content, APT has a mechanical strength two times higher than that of MPT. APT and MPT have comparable whiteness.

4.   MPT has dominated the porcelain tile market to date, and its in-process behavior is better understood compared to that of APT. However, APT represents a promising option for replacing, or for use in combination with, MPT on a large scale, in order to achieve better results.

**Author Contributions:** Conceptualization, K.L. and A.D.N.J.; writing—original draft preparation, K.L., E.d.S.C. and A.D.N.J.; data collection, E.d.S.C.; writing—review and editing, K.L., E.d.S.C. and A.D.N.J.; supervision, A.D.N.J.; project administration, K.L.; funding acquisition, K.L. and A.D.N.J. All authors have read and agreed to the published version of the manuscript.

**Funding:** This research was funded by the Department of Science and Technology of Shaanxi Province, grant number 2020GY-314; Shaanxi University of Technology, grant number SLGRCQD2021; and the Brazilian National Council of Scientific and Technological Development CNPQ, grant number 302555/2020-0.

**Acknowledgments:** The authors gratefully acknowledge the financial support provided above mentioned institutions.

**Conflicts of Interest:** The authors declare no conflict of interest.

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
