# Peer review of "Comparison between Mullite-Based and Anorthite-Based Porcelain Tiles: A Review"

_2673-4117, doi:10.3390/eng4030123_

Round 1

Reviewer 1 Report

Manuscript Number: eng-2509201

The manuscript “Comparison between mullite-based and anorthite-based porcelain tiles: a review” presents literature review about chemical composition, processing and characteristic of mullite-based and anorthite-based porcelain tiles. The topic is suitable for this journal. However, the manuscript shows some deficiencies:

  1. Lines 37-39: provide literature source for this claim (depletion of clay minerals)
  2. Lines 40-42: different polymorphs of pure TiO2 are white, thus cannot produce dark color.
  3. Lines 56-60: cite the appropriate literature sources for given data
  4. Lines 90-96: cite the appropriate literature sources for given data
  5. Lines 125-17: cite the appropriate literature sources for given data
  6. Lines 130-135: for given conclusion authors should either show data or cite appropriate literature; refereeing to text in section 2.1 is not sufficient
  7. Lines 146-150; 154-156: 157-165;188-194; 199-204; 267-280; 285-292; 499-505; 540-544- cite the appropriate literature sources for given discussion
  8. Lines 206-207: insert “of green bodies” after chemical composition
  9. Line 210: cite literature source
  10. Lines 212-215: repetition, should be deleted
  11. Lines 224-226: it is not clear if author discus their own data or literature
  12. Table 2: it is not clear which values are from which references 9some values are without reference)

Starting from section 5.2. it is not certain if there are problems with the pdf creation or with the submitted manuscript, so please have this in mind when reading following comments:

  1. Subtitle 5.2 refers only to APT, yet in the text there is continuous comparison with MPT. Unclear.
  2. Figure 3: it is unclear if shown data are original work from the authors or taken from the literature. That needs to be clarified, also additional description if needed either detailed description of sample preparation in the case of original data, or criteria why is this example chosen from the literature.
  3. Lines 530-539: ?
  4. Lines 560-586: identical text as at lines 507-529 – that needs to be corrected
  5. Table 3: clearly mark in the Table and describe in the text what is original work from Authors and what is from the literature

Having all this in mind I can recommend this manuscript to be published in Eng after minor revision.

General comment: The language in which the manuscript is written needs to be improved thoroughly.

Reviewer 2 Report

The text should be checked carefully. There are many inaccuracies.

Some comments.

1. Part 5 should be called “Mechanical behavior” and not “Phase evolution”.

2. Why in Table 2 references are given only for some values? Uniformity is required.

3. Lines 378-498 are empty

4. Lines 529 – 539 look strange and needs corrections

5. There are a lot of repetitions:

-        - Lines 332 and 343

-        - Lines 509-510 and 562-563

-        - Lines 514-515 and 571-572

6. To check forming pressure in Table 3 (35-45 MPa) and in Line 602 (33-45 MPa)

Reviewer 3 Report

The article by Kun Li , Eloise Cordeiro , Agenor De Noni Junior presents a comparison of porcelain stoneware based on mullite and anorthite. The features of the composition of porcelain stoneware based on mullite and on the basis of anorthite, the difference in sintering modes and the resulting consumer properties are described. The conclusion is made about the possible applications of each of the materials. The article is well written, but contains little illustration and comparison of materials. As you read the manuscript, it is difficult to keep such a large amount of information in mind. The authors should break the material into blocks and make a comparison for each of the parameters with the addition of graphics. I think these changes will improve the article.

Round 2

Reviewer 3 Report

The article can be published in this form